# Reconstruction of Continuous High-Resolution Sea Surface Temperature Data Using Time-Aware Implicit Neural Representation

Yang Wang [1,*] , Hassan A. Karimi [1] and Xiaowei Jia [2]

1    Geoinformatics Laboratory, School of Computing and Information, University of Pittsburgh, 135 N Bellfield Ave, Pittsburgh, PA 15213, USA; hkarimi@pitt.edu
2    Department of Computer Science, School of Computing and Information, University of Pittsburgh, Pittsburgh, PA 15213, USA; xiaowei@pitt.edu
*    Correspondence: yaw70@pitt.edu; Tel.: +1-412-708-6722

**Abstract:** Accurate climate data at fine spatial resolution are essential for scientific research and the development and planning of crucial social systems, such as energy and agriculture. Among them, sea surface temperature plays a critical role as the associated El Niño–Southern Oscillation (ENSO) is considered a significant signal of the global interannual climate system. In this paper, we propose an implicit neural representation-based interpolation method with temporal information ($T\_INRI$) to reconstruct climate data of high spatial resolution, with sea surface temperature as the research object. Traditional deep learning models for generating high-resolution climate data are only applicable to fixed-resolution enhancement scales. In contrast, the proposed $T\_INRI$ method is not limited to the enhancement scale provided during the training process and its results indicate that it can enhance low-resolution input by arbitrary scale. Additionally, we discuss the impact of temporal information on the generation of high-resolution climate data, specifically, the influence of the month from which the low-resolution sea surface temperature data are obtained. Our experimental results indicate that $T\_INRI$ is advantageous over traditional interpolation methods under different enhancement scales, and the temporal information can improve $T\_INRI$ performance for a different calendar month. We also examined the potential capability of $T\_INRI$ in recovering missing grid value. These results demonstrate that the proposed $T\_INRI$ is a promising method for generating high-resolution climate data and has significant implications for climate research and related applications.

**Keywords:** deep learning; implicit neural representation; sea surface temperature; super-resolution; satellite retrieval climate data; temporal information

## 1. Introduction

El Niño–Southern Oscillation (ENSO) is the strongest signal of interannual variability in the climate system. Research on ENSO is crucial for understanding the complex interactions between the oceans, atmosphere, and climate, as well as for developing strategies to mitigate the potential impacts of climate variability and change, because it not only directly causes extreme weather events such as droughts in the tropical Pacific and its surrounding areas but also indirectly affects the weather and climate in other parts of the world through teleconnections, triggering meteorological disasters [1–3]. Currently, many studies have been conducted to investigate the teleconnections of ENSO and its predictive capabilities regarding other climate factors [4–7]. In recent years, related research has gone beyond the use of scalar indices such as NINO 3.4 to characterize ENSO. The limitations of scalar indices in capturing the full complexity of ENSO dynamics have spurred interest in alternative methods for describing ENSO events. For example, some studies have used empirical orthogonal functions of SST in the Pacific Ocean as indices, rather than NINO3.4 [8]. Others have employed a non-homogeneous hidden Markov model (NHMM)

to simulate the monthly scale tropical Pacific SST and defined five different hidden states for each month [9,10].

Commonly used SST data are climate model outputs or satellite-derived data. The accuracy and richness of information in these grid-type representations are controlled by their resolution. Not only SST data, high-resolution (HR) climate data are of great significance for climate simulations, agriculture, and other fields. Traditional methods for obtaining HR climate data mainly rely on interpolation, such as bicubic and bilinear interpolation [11]. These methods do not require training data but often result in over-smoothed outcomes. With the development of deep learning techniques, an increasing number of deep learning models have been applied to generate HR climate data and address the issue of over-smooth in the process. For example, Vandal et al. [12] used super-resolution convolutional neural networks (SRCNN) to generate HR rainfall data. Ducournau et al. [13] also used SRCNN to increase the resolution of satellite-derived SST data. Stengel et al. [14], based on SRGAN, enhanced the resolution of wind and solar data by a factor of 50. Wang et al. [15] explored the applicability of SinGAN in generating HR climate data. However, a common limitation of these models is that their enhancement scales are fixed. Once trained, they can only be used for reconstructing HR data at a pre-defined gridded structure, whereas climate analysis often requires HR data at arbitrary locations over different scales.

To overcome this limitation, we propose to develop a new approach for continuous HR data reconstruction based on the implicit neural representation (INP) method [16,17]. The idea is to represent an object as a multi-layer perceptron (MLP) that maps coordinates to signals. It has been extensively used in 3D tasks, such as simulating 3D surfaces [18,19] or structures [20,21]. INP is also widely used in research on image reconstruction [22–24]. One improvement to this approach is to use an encoder to construct a shared feature space that can be used to represent every sample with an implicit representation [25,26]. For example, Chen et al. [27] constructed a continuous image representation using implicit neural representation to overcome the limitation of implicit sample-specific neural representation.

In this study, we introduce an interpolation method that integrates temporal information based on implicit neural representation ($T\_INRI$). By fusing deep learning models with an interpolation technique, this method can produce HR climate data at arbitrary scales. After the model's training, our method is capable of not only enhancing the resolution based on the scales used during training but also enhancing the resolution of climate data based on scales not seen in the training phase. While implicit neural representations have proven successful in 3D and image tasks, their application in climate research has yet to be explored. This paper addresses this gap by focusing on a specific use case: reconstructing HR SST data from low-resolution (LR) samples. We consider the reconstruction of HR SST data as a problem of estimating unknown values (HR sample positions) based on known values (LR sample positions). Within the proposed interpolation approach, LR SST samples undergo an encoding process to attain corresponding implicit neural representations for each grid cell. These representations act as anchors for predicting the values at positions in the HR result. The method establishes a relationship between the implicit neural representations at these anchor sites and unknown positions, factoring in grid size, central point coordinates, and other parameters. A decoder then infers the values at these unknown positions, yielding a comprehensive HR climate data result. Additionally, we incorporate temporal information, specifically the calendar month during which samples are acquired. The model employs one-hot encoding to capture the sample's calendar month and leverages a learnable matrix to enhance its capability to process intricate temporal information. Our results suggest that the proposed interpolation method outperforms conventional interpolation methods at different enhancement scales. Compared to CNN-based approaches for generating HR climate data, the proposed method's advantage lies in its flexibility. Comparisons between results with and without temporal information underscore the merit of incorporating temporal information, as it elevates the quality of the

generated HR SST data. Additionally, we explore the method's potential to recover missing data by setting the enchantment scale to 1.

The paper is organized as follows. Section 2 describes the data used in the study and Section 3 introduces the details of the proposed *T_INRI* method and the network setup. Section 4 discusses the results. Section 5 gives conclusions and discusses future research directions.

## 2. Dataset

We use the GHRSST Level 4 MUR Global Foundation Sea Surface Temperature Analysis dataset (MUR SST) in our experiments [28]. MUR SST is one of the current highest-resolution SST analysis datasets, providing global daily SST from 31 May 2002 to the present. The spatial resolution of MUR SST is $0.01° \times 0.01°$, roughly at 1 km intervals. MUR SST combines three types of satellite SST datasets: HR infrared SST data of about 1 km, medium-resolution AVHRR (infrared) SST data of 4 to 8.8 km, and microwave SST data with a sampling interval of 25 km [29]. In addition to these three satellite-derived data, MUR also assimilates two types of in situ SST measurements to enhance the estimation of the underlying temperature. Due to the computational burden of using the entire dataset, we extracted the SST in the region 180°W–90°W and 5°S–5°N as the study area, as shown in Figure 1. The training dataset used for model development comprises daily samples spanning from 1 June 2002 to 31 December 2016. Each sample consists of $1002 \times 9001$ cells, resulting in a total of 5398 samples, while the daily data from 1 January 2017 to 31 December 2022 with a total of 2188 samples serve as the validation dataset. For testing, in addition to the MUR SST validation dataset described above, we also report results for the MUR SST Monthly Mean dataset as an external validation. MUR SST Monthly Mean dataset is created by NOAA NMFS SWFSC ERD based on the GHRSST Level 4 MUR SST daily dataset mentioned above. It has the same spatial resolution as daily data. We use the same study area as the one used for the MUR SST dataset. The MUR SST Monthly Mean dataset is comprised of monthly samples spanning from 2003 to 2020.

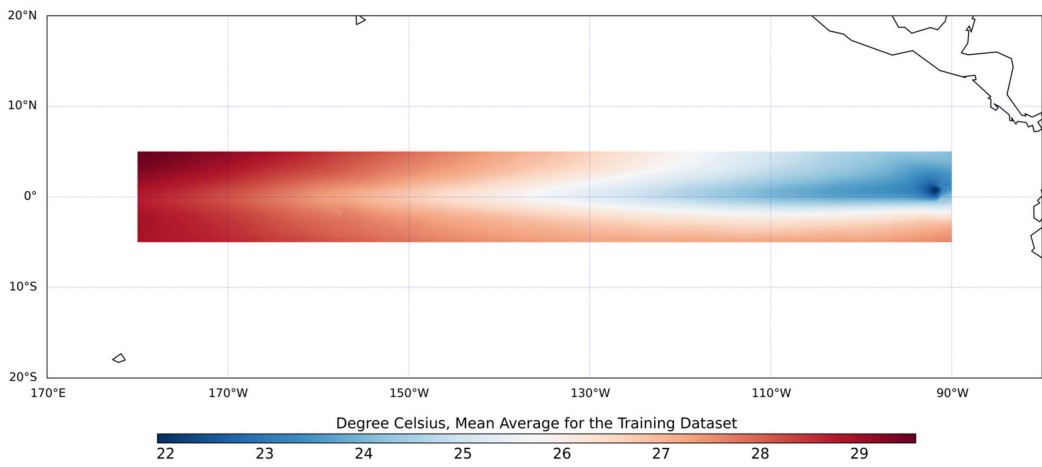

**Figure 1.** Region of SST data retrieval for the study area.

## 3. Method

### 3.1. Overview of Proposed Method

The essence of obtaining HR climate data using an interpolation-based method is to derive the climate values of HR grid cells based on the values of LR grid cells. Traditional spatial interpolation methods are based on information from the original data domain. For example, bicubic interpolation computes the value of an unknown grid cell by taking the weighted average of the surrounding 16 grid cells. In our proposed method, instead of directly interpolating from the original data domain, the interpolation occurs in the implicit representation domain. Essentially, the model is comprised of two parts: building implicit representation domain and interpolation. We first introduce the two deep learning

models employed in this method and then provide a detailed description of the specific steps within the proposed *T_INRI* method.

### 3.2. Enhanced Deep Super-Resolution Network and Multilayer Perceptron

The proposed method integrates two distinct deep learning architectures. The first architecture is from the enhanced deep super-resolution network (EDSR) [30], and we use the initial segment before the upsampling layer, as depicted in Figure 2A. Within our method, EDSR is constructed with a convolutional layer, succeeded by 16 residual blocks. The addition of multiple residual blocks equips the network with the capability to discern complex patterns from the training process. Each of these blocks consists of convolutional layers, succeeded by a ReLU activation function. The essence of these residual blocks is to ascertain the residual between the input and output, eschewing direct output learning. This design implies that the network predominantly learns variations from identity mapping, promoting training stability. We utilize this structure as an encoder to transition the input data from its original data domain to an implicit representation domain. Rather than modifying the spatial dimensions of the input, the encoder enhances the depth at each location, yielding this implicit representation. The second architecture is a five-layer multilayer perceptron (MLP), as depicted in Figure 2B. Each hidden layer processes inputs from its preceding layer, undergoes a weighted summation, and produces outputs via the ReLU activation function. In our design, the hidden layers possess a dimensionality of 256, with the output layer dimensioned at 1. This MLP, in our method, functions as a decoder, determining the value at a specific location based on relevant input information.

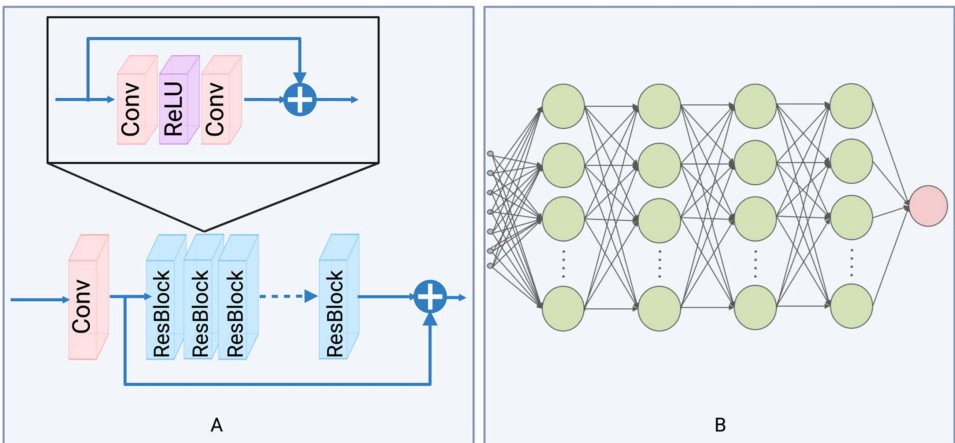

**Figure 2.** The main architecture of the network used in the proposed method: (**A**) a partial structure of EDSR, serving as the encoder in the proposed interpolation method; (**B**) a multi-layered MLP, functioning as the decoder in the proposed interpolation method.

### 3.3. Implicit Neural-Representation-Based Interpolation with Temporal Information
#### 3.3.1. Implicit Neural-Representation-Based Interpolation

An implicit representation domain is achieved through the utilization of a convolutional neural network-based encoder, which outputs the feature map and retains the same size as the input sample. At each location in the input sample, the corresponding feature vector constitutes the implicit representation of that location. Each grid cell can be represented by its center position. We hypothesize that the value of each grid cell in the sample can be obtained by inputting the grid cell's implicit representation, coordinates, and size into a decoder:

$$V = D_\partial(z, [p_1, p_2]) \tag{1}$$

where $z$ represents the implicit representation of the target position, and $[p_1, p_2]$ represents the coordinates of the center point of the grid. $D_\partial$ is the decoder, which is the MLP as described in Section 3.2. We assume that the implicit representation is uniformly distributed

in the feature map domain, so the value of the grid cell at an arbitrary location in the domain can be obtained through the following function $D$:

$$v_{unknown} = D(z^*, \Delta d, [s_1, s_2], [x_1, x_2]) \tag{2}$$

where $z^*$ is the implicit representation of the nearest known grid to the predicted unknown grid, $[s_1, s_2]$ represents the length and width of the unknown grid, $[x_1, x_2]$ represents the coordinates of the center of the unknown grid, and $\Delta d$ is the Euclidean distance between the two grids, which is calculated by:

$$\Delta d = \sqrt{(x_1 - p_1)^2 + (x_2 - p_2)^2} \tag{3}$$

The model represents the length and width of the grid through $s_1$ and $s_2$, thus providing the information of grid size. We take grid size into consideration because grids of different sizes might share the same central point. However, grids of varying sizes often represent distinct values due to the differences in the areas they cover. By incorporating information about grid size, we enhance the decoder's ability to differentiate situations with the same grid center but differing resolutions. For the HR climate data projection task, the implicit representation of each grid cell in the LR input data can be obtained through a shared encoder among all samples. Every grid cell with implicit representation in the LR input can be used as an anchor point for generating the corresponding HR data. In Figure 3, blue grid cells represent LR input data, and the values of red grid cells which represent one grid cell of HR data can be obtained by the nearest anchor points $Anchor_{bl}$ and Equation (2).

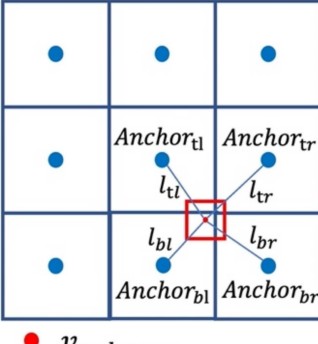

$\bullet$ $v_{unknown}$

**Figure 3.** Relative position of unknown grid cell in HR data and anchors in LR input.

However, based on the First Law of Geography, the closer the points are in space, the higher the probability they have similar feature values, and the farther the points are in space, the lower the probability they have similar feature values. For any unknown grid cell, we can compute its value in the latent feature domain based on its nearest grid cell at the coarse resolution. However, this way remains limited as it ignores the valuable information from other surrounding known grids. Furthermore, just using one nearest anchor point can result in discontinuous patterns in the generated HR output because the anchor point (i.e., the nearest grid cell) might abruptly shift from one to another when the target grid cell moves gradually over space. The proposed method eliminates the discontinuity of the result through the weighted average of interpolated results from multiple anchor points around the unknown grid cell, where the weights are based on the distance between anchor points and the target grid cell.

We consider the four points around the unknown grid as anchor points. As seen in Figure 3, they are located at points $Anchor_{tl}$, $Anchor_{tr}$, $Anchor_{bl}$, and $Anchor_{br}$, respectively, on the upper left, upper right, lower left, and lower right of the unknown grid cell. *T_INRI* first obtains the interpolation results $v_{unknown_i}$ based on each anchor point for the unknown

grid through Equation (2). Then, the final prediction value of the unknown grid is obtained by weighing the average of the four results according to the inverse distance.

$$v_{unknown} = \sum_{i \in (br,bl,tr,tl)} W_i * v_{unknown\_i} \qquad (4)$$

$$W_i = \frac{\frac{1}{d_i}}{\sum_{i \in (br,bl,tr,tl)} \frac{1}{d_i}} \qquad (5)$$

where $d_i$ represents the distance between the center of the unknown grid cell and each of the four anchor points.

### 3.3.2. Temporal Information Embedding

The process of getting HR data from the LR counterpart is commonly considered to be an ill-posed problem, as one LR pattern can correspond to multiple different HR patterns, and the model's training process tries to average all plausible solutions. In this LR to HR projection process, additional information is considered to be useful, as it helps the model choose the most appropriate HR pattern among all plausible solutions. For example, in the case of climate data such as sea surface temperature, temporal information, i.e., the time period the sample was taken from, can be utilized. Climate variables may exhibit a relatively uniform spatial distribution at one time period and a non-uniform distribution at another time period, and these two different HR data may correspond to the same LR sample. Adding temporal labels helps the model lock onto the most appropriate HR solution. In our research, the calendar month from which an SST sample was taken was considered as temporal information. Our hypothesis is that HR SST samples from different months (e.g., January and July) can correspond to the same LR sample. By providing the LR sample's calendar month information, the model can more robustly find the corresponding HR pattern by using implicit neural representation.

In order to handle temporal information, the calendar month of each sample is first encoded as a 12-dimensional one-hot vector. The vector consists of 0 s in all cells with only one cell with a value of 1 uniquely identifying the calendar month of the sample. This one-hot vector is then projected into a 12 dimensional embedding space by multiplying it with a learned parameter matrix $W_p$ as follows:

$$T_{label} = W_p \times t \qquad (6)$$

where $T_{label} \in \mathbb{R}^{12 \times 1}$ is a vector representing the temporal information of a sample, $W_p \in \mathbb{R}^{12 \times 12}$ is a learnable matrix, and $t \in \mathbb{R}^{12 \times 1}$ is a one-hot vector. The updated version of Equation (2) is as follows:

$$v_{unknown} = D(z^*, \Delta d, [s_1, s_2], [x_1, x_2], T_{label}) \qquad (7)$$

### 3.4. Training and Setup

In this section, we discuss the training procedure of *T_INIRI*. Figure 4 shows how *T_INIRI* works and its training steps. Our main goal during training is to estimate the parameters of the encoder and the MLP using our training data. For each LR input, the encoder provides an implicit representation for every grid in the LR sample. For every unknown grid in the HR data, our method looks for the four closest known points used as anchors. Using the MLP, we then figure out the value of this unknown grid. After repeating this for every unknown grid in the HR, we get the HR result for the given LR input. As previously discussed, we expect the model to be used to obtain HR data at any enhancement factor after training.

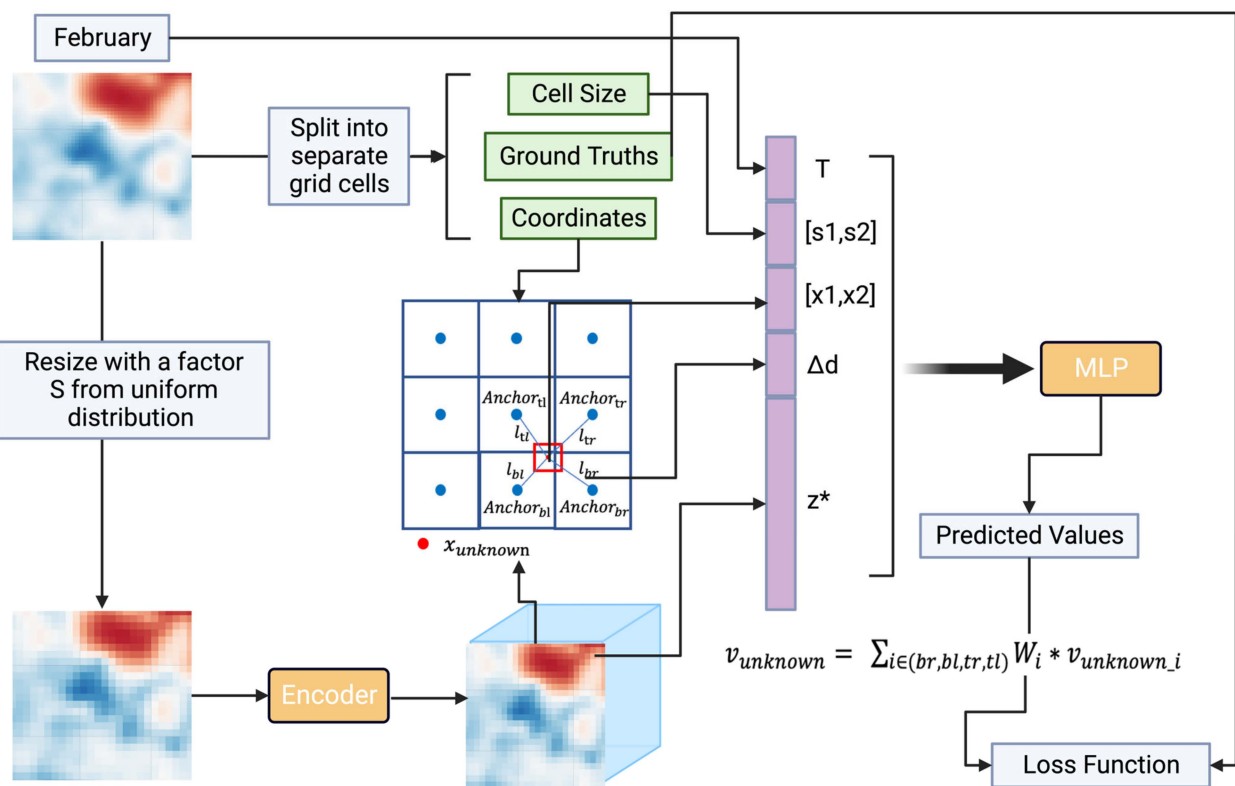

**Figure 4.** Workflow and training steps of the proposed interpolation method.

The training process of the model employs self-supervised learning. For each HR input sample, the enhancement factor *k* is randomly selected from a uniform distribution from 1 to 5. We follow the previous research on the encoder and use a $48 \times 48$ patch as the input size for the encoder during the training process. This way, for the HR input sample, the model first cuts out a $48k \times 48k$ size patch. The patch is divided into separate grid cells as unknown cells. Each grid cell contains the grid value as the ground truth, the length and width of the grid as the grid size, the center point coordinates of the grid as the target coordinates, and the temporal information of the grid (for the grid cells on the same sample, their temporal information is consistent). The $48k \times 48k$ size patch is down-scaled to a $48 \times 48$ LR data by a factor of *k* using nearest neighbor interpolation. The LR data are then input into the encoder to obtain a feature map with size $(64 \times 48 \times 48)$, where 64 is the length of implicit representation. After we get the implicit representation for each known grid in the LR input, the proposed method searches for the nearest four anchors for each target coordinate based on the feature map. The proposed method calculates the value of the target grid for each of the four anchors and then takes the IDW-weighted average to get the final target grid value. After searching and computing for every unknown grid in the HR sample, our method combines the results of all these unknown grids to form the generated HR result. This result is then compared with the ground truth to calculate the loss. Based on this loss, the parameters of both the encoder and the MLP are updated.

We use Charbonnier Loss as the loss function for the optimization [31,32]. The batch size used in our experiment is 16. The training is performed for a total of 1000 epochs, with the initial learning rate set to 0.0001. The learning rate is adjusted after every 200 epochs, decreasing by a factor of 0.5.

### 3.5. Validation

We conducted an evaluation on multiple scales, including the scales used for training and higher spatial resolutions. For the MUR SST dataset and SST Monthly Mean dataset, we evaluated the performance of the models on out-of-training enhancement scales of 8, 12, 14, 16, and 20, in addition to the in-training enhancement scales of 2 to 5. We used

the original resolution MUR SST and SST Monthly Mean dataset as the ground truth. For each enhancement scale *k*, we first down-sampled the original resolution samples to the corresponding LR inputs using the nearest neighbour interpolation by a scale of ($1/k$) and then applied different models to increase the resolution to the original resolution and compared with the ground truth. Root mean squared error (RMSE) between ground truth HR data and generated HR data was used to evaluate performance.

## 4. Results and Discussion

### 4.1. Comparison of Results Generated by Different Models for Arbitrary Scale HR

To verify the effectiveness of the proposed model in generating HR SST data of different scales, we compared the results obtained by the proposed $T\_INRI$ with those obtained by other interpolation methods, including two traditional interpolation methods, bicubic interpolation and bilinear interpolation, as well as two deep learning models, SRCNN and SRGAN. As mentioned previously, SRCNN and SRGAN were used to improve resolution with a fixed enhancement scale. Once trained, they cannot be used for arbitrary multiple-resolution enhancements. In our results, we only compared two enhancement scales, 4 and 8 for SRCNN and SRGAN. The training process of SRCNN follows [12], and the training process of SRGAN follows [14]. For each enhancement scale, the original resolution serves as the ground truth. We down-sampled the original data to the corresponding size as LR input for each scale using an average pooling, respectively. To analyze the impact of temporal information on the proposed interpolation method, we also trained a decoder and MLP for interpolation without incorporating time information. The method without embedded temporal information is denoted as $INRI$.

The results of the comparison are given in Table 1. We not only emphasize the enhancement scales observed during the training phase, specifically scales 2 to 5, but also consider the scales beyond training scales, termed as "out-of-training scale" in the table. The table also delineates results for both versions: one with temporal information and one without. It is evident from the data that across various enhancement scales, the performance of the proposed method, irrespective of with or without temporal information, markedly surpasses that of traditional interpolation methods. For instance, when the enhancement scale is 2, both bicubic and bilinear have an RMSE of 0.014 and the RMSE corresponding to $T\_INRI$ achieves merely 0.004, which is a 71% improvement compared to the previous two methods. When the enhancement scale is 5, the RMSE for Bilinear is 0.050, while for the same scale, $T\_INRI$'s RMSE is 0.019, denoting a 62% enhancement. For the out-of-training scale, the superiority of $T\_INRI$ remains evident. Specifically, when the enhancement scales were 8, 10, 12, 14, 16, and 20, respectively, the HR results produced by $T\_INRI$ exhibit approximately 53%, 49%, 45%, 50%, 42%, and 39% enhancement compared to Bilinear. Unlike traditional bicubic and bilinear methods that work solely on original values, $T\_INRI$ operates on a deeper level of feature domain [33]. When $T\_INRI$ is used to upscale an LR input, it does not just interpolate between the original data domains; it predicts HR details based on its learned understanding of how HR should be enhanced. This leads to more details, which is particularly noticeable in textures and edges. Through its deep learning architecture, it can adapt its upscaling process based on different areas of input data. This adaptability allows for more nuanced detail preservation as opposed to the one-size-fits-all approach of bicubic or bilinear methods [27,34].

For SRCNN and SRGAN, when the enhancement scale is 4, the results from SRCNN and SRGAN are close to those of $T\_INRI$. However, when the enhancement scale is 8, SRCNN and SRGAN slightly outperform $T\_INRI$. As previously discussed, the architectures of these two models are specifically designed and trained for a certain enhancement scale; hence, it is anticipated that their performance under specific resolution conditions would be comparable to that of $T\_INRI$. More intricate model structures can be formulated based on SRCNN and SRGAN to boost the performance of the HR outputs. However, their network architectures and parameters are defined for specific enhancement scales, making it infeasible to compare them with other models under arbitrary enhancement conditions.

**Table 1.** Quantitative comparison of different methods on MUR SST validation dataset.

| Method | In-Training-Scale RMSE (°C) | | | | | Out-of-Training-Scale RMSE (°C) | | | | |
|---|---|---|---|---|---|---|---|---|---|---|
| | ×2 | ×3 | ×4 | ×5 | ×8 | ×10 | ×12 | ×14 | ×16 | ×20 |
| Bicubic | 0.014 | 0.027 | 0.040 | 0.051 | 0.082 | 0.099 | 0.112 | 0.129 | 0.135 | 0.154 |
| Bilinear | 0.014 | 0.027 | 0.039 | 0.050 | 0.079 | 0.095 | 0.107 | 0.124 | 0.130 | 0.148 |
| SRCNN | - | - | 0.015 | - | 0.035 | - | - | - | - | - |
| SRGAN | - | - | 0.014 | - | 0.033 | - | - | - | - | - |
| *T_INRI* | 0.004 | 0.009 | 0.014 | 0.019 | 0.037 | 0.048 | 0.058 | 0.063 | 0.075 | 0.090 |
| *INRI* | 0.013 | 0.015 | 0.017 | 0.021 | 0.038 | 0.049 | 0.059 | 0.065 | 0.077 | 0.092 |

From the performance comparison between *T_INRI* and *INRI* across varying enhancement scales, the beneficial impact of temporal information on the results becomes evident. For in-training scales, the advantage conferred by temporal information tends to diminish as the scale increases. For instance, at an enhancement scale of 2, *INRI* registers an RMSE of 0.013 °C, aligning with the results from bicubic and bilinear methods. The RMSE of 0.004 °C demonstrated by *T_INRI* underscores the substantial benefits derived from incorporating temporal information. As the enhancement scale progresses from 3 to 5, the improvements observed with AW are 0.006 °C, 0.003 °C, and 0.002 °C, respectively, indicating a declining trend. Considering out-of-training scales, *T_INRI* consistently outperforms *INRI*, albeit with a slight margin, and this superiority remains stable across different enhancement scales. Enhancing the resolution of LR input by a significant scale is inherently challenging, particularly for scales not encountered during training. Under such circumstances, the influence of temporal information on the model is relatively diminished.

The qualitative comparison results are illustrated in Figures 5 and 6. To better showcase the discrepancies in the HR results generated by different models, in Figure 5, we present the results for various models when the enhancement scale is set to 2, and these results are derived based on the RMSE calculated per cell. In Figure 6, we display the outcomes obtained by subtracting the HR results generated by different models from the ground truth for a sample. As can be seen from Figure 5, when the enhancement scale is set to 2, the spatial distribution of *T_INRI*'s results aligns closely with the ground truth. Observing the top-left figure reveals that most regions are blue, indicating that the difference between *T_INRI*'s output and the ground truth is nearly zero. The error distribution patterns of *INRI*, bicubic, and bilinear are similar, corroborating the results presented in Table 1. Comparing *INRI* and *T_INRI* highlights the significant advantages that temporal information offers in enhancing the quality of the resulting HR output, where *INRI* exhibits higher RMSE values in the northern and central regions of the area. However, when the enhancement scale is set to 5, as depicted in Figure 6, despite the observation that the proposed models with and without temporal information still outperform bicubic and bilinear in terms of spatial error distribution, the influence of spatial information on the proposed models becomes marginal. The spatial distributions between the two are largely consistent across most regions. We also find that traditional interpolation models tend to overestimate SST in areas where *INRI* and *T_INRI* typically underestimate, and vice versa. These differences may arise because traditional methods interpolate in the original data space, while the proposed interpolation takes place in the implicit neural space.

To further validate the robustness and applicability of the proposed method, an additional dataset was utilized. Specifically, the MUR monthly mean SST dataset was employed, which shares an identical spatial resolution with the MUR daily dataset. For evaluation purposes, the original spatial resolution was treated as the ground truth. Corresponding LR inputs were derived using average pooling across different scales. From Table 2, we observe the results based on the additional dataset. The proposed interpolation method, which embeds temporal information, consistently exhibits superior performance across various enhancement scales. When comparing results with and without temporal information, it is evident that for the MUR monthly mean SST dataset, the embedding of temporal

information contributes to a more substantial performance improvement compared to the previous validation dataset. For instance, at an enhancement scale of 5, the improvement between with and without temporal information in the additional dataset is 0.008 °C. This gap narrows to 0.004 °C at an enhancement scale of 16. In contrast, for the same enhancement scales, the validation dataset shows a consistent improvement of only 0.002 °C. This indicates that embedding temporal information enhances the generalization capability of the proposed interpolation method.

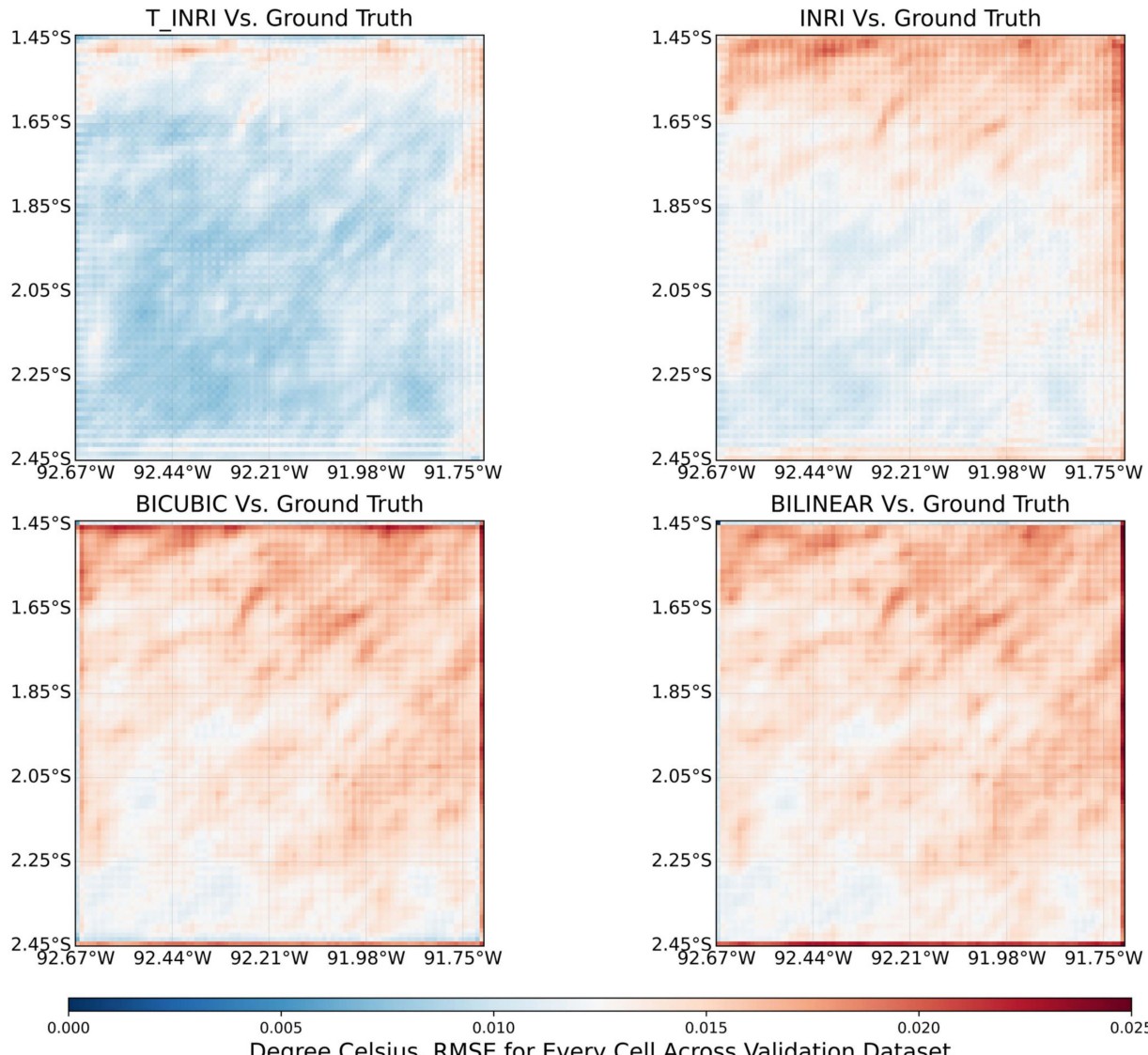

**Figure 5.** Qualitative comparison of generated HR SST data from different models. RMSE is calculated for every cell across validation dataset. The enhancement scale is ×2.

**Table 2.** Quantitative comparison of different methods on MUR SST monthly dataset.

| Method | In-Training-Scale RMSE (°C) | | | | | Out-of-Training-Scale RMSE (°C) | | | | |
|---|---|---|---|---|---|---|---|---|---|---|
| | ×2 | ×3 | ×4 | ×5 | ×8 | ×10 | ×12 | ×14 | ×16 | ×20 |
| Bicubic | 0.014 | 0.028 | 0.041 | 0.053 | 0.084 | 0.100 | 0.115 | 0.128 | 0.138 | 0.157 |
| Bilinear | 0.014 | 0.028 | 0.040 | 0.052 | 0.081 | 0.097 | 0.111 | 0.123 | 0.132 | 0.151 |
| *T_INRI* | 0.005 | 0.010 | 0.015 | 0.020 | 0.038 | 0.050 | 0.060 | 0.069 | 0.077 | 0.092 |
| *INRI* | 0.014 | 0.021 | 0.023 | 0.028 | 0.044 | 0.055 | 0.064 | 0.073 | 0.081 | 0.096 |

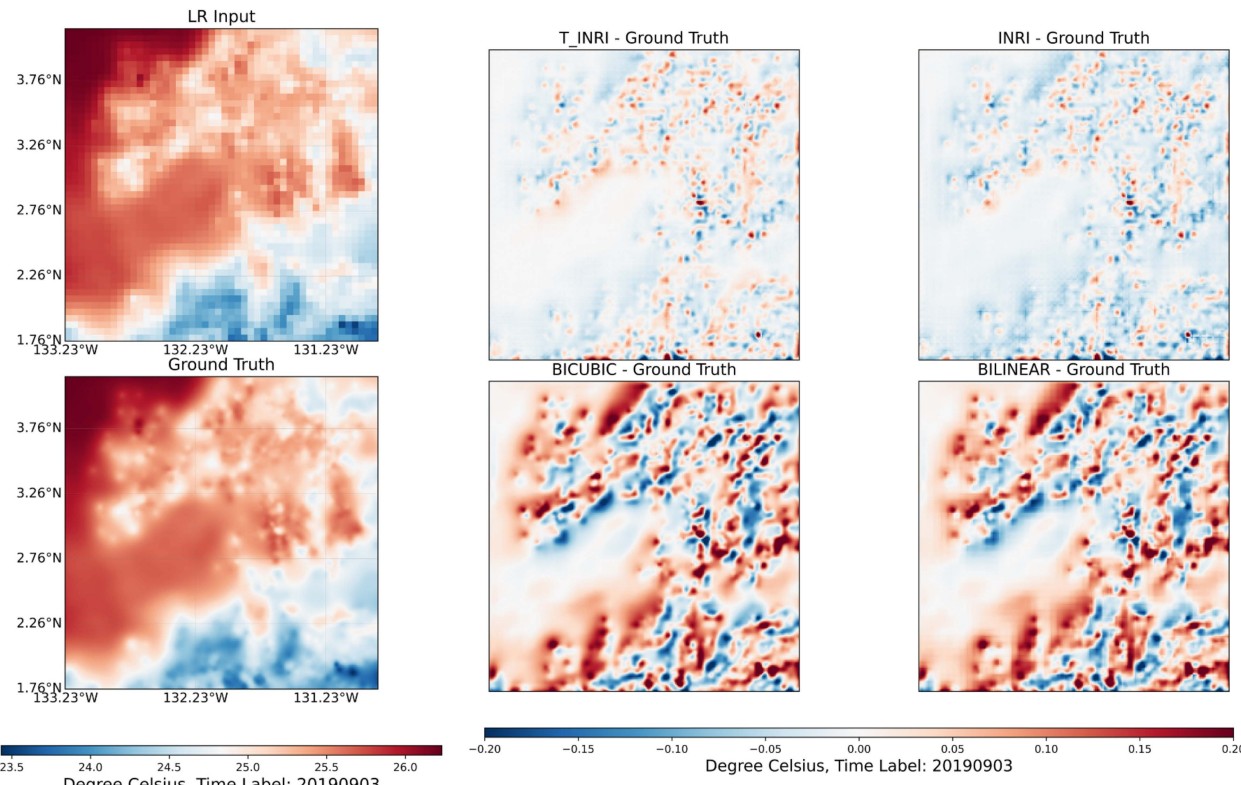

**Figure 6.** Qualitative comparison of generated HR SST data from different models. The sample is from the MUR SST validation dataset with time label 20170924. The enhancement scale is ×5.

### 4.2. Analysis of the Impact of Temporal Information

To further investigate the influence of temporal information on the results, we compared the performance with and without temporal information embedded. Figure 7 depicts the error density between HR data generated based on different calendar months and the ground truth observations, with an enhancement scale set to 3. Notably, results incorporating temporal information consistently exhibit smaller errors than those without, and this improvement remains stable across different calendar months. The difference in average monthly error between the two approaches is approximately 0.005 °C. Comparing results across various months, December manifests the smallest average error at 0.009 °C. Meanwhile, August has the lowest standard deviation in error, amounting to 0.003 °C. For results without temporal information, December similarly yields the smallest average error, recorded at 0.013 °C, while July has the lowest error standard deviation, standing at 0.004 °C.

In order to better illustrate the impact of temporal information, we present in Figure 8 the improvement in results obtained with temporal information at different scales and seasons. It is apparent that the figure reflects the same trend observed in Table 1. Specifically, as the enhancement scale increases (here represented as 2, 3, 4, and 8), the difference between results with and without temporal information diminishes. This diminishing trend with respect to the enhancement scale is consistent across each calendar month. For instance, at an enhancement scale of 3, the average difference between results with and without temporal information for every calendar month is approximately 0.005 °C. When the enhancement scale is increased to 8, this average difference narrows to about 0.003 °C. From the analysis of the impact of temporal information, we can conclude that incorporating temporal data can enhance the HR SST results obtained for each calendar month. However, as the enhancement scale increases, the positive influence of temporal information consistently diminishes for every month.

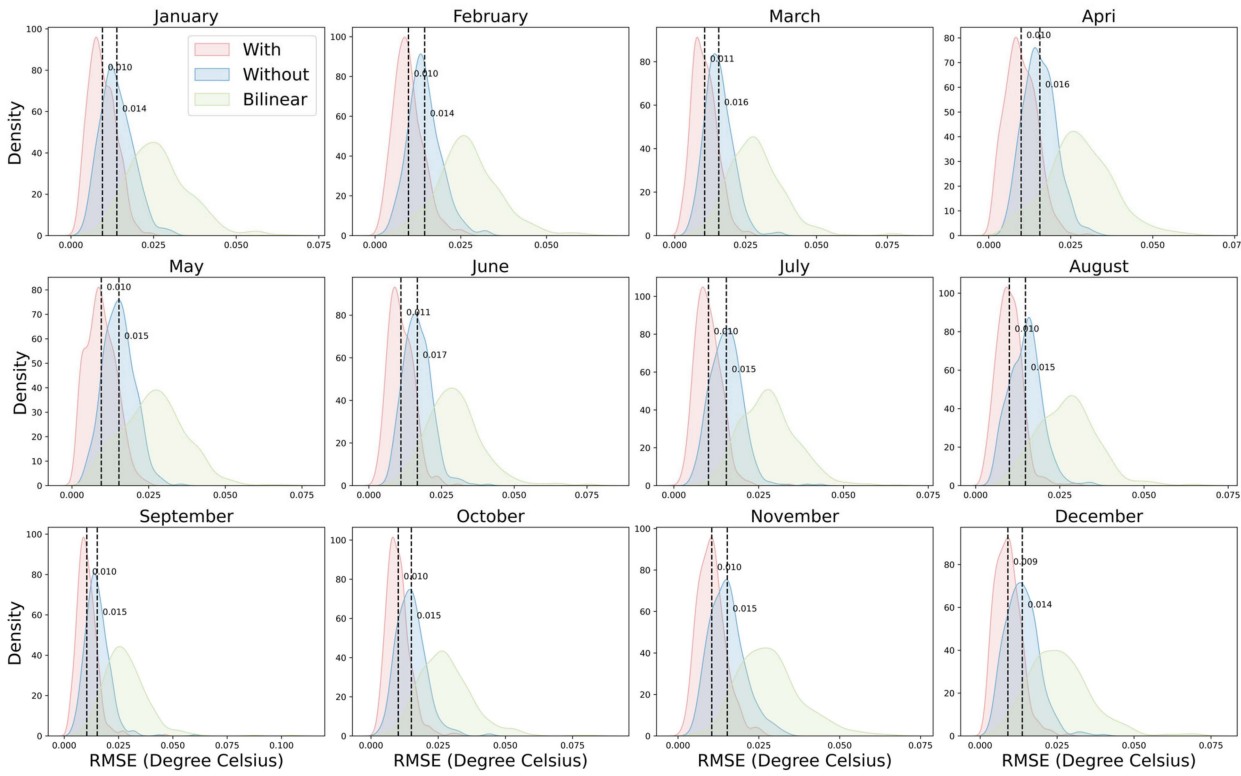

**Figure 7.** Density plot of error between with and without temporal information. The enhancement scale is 3.

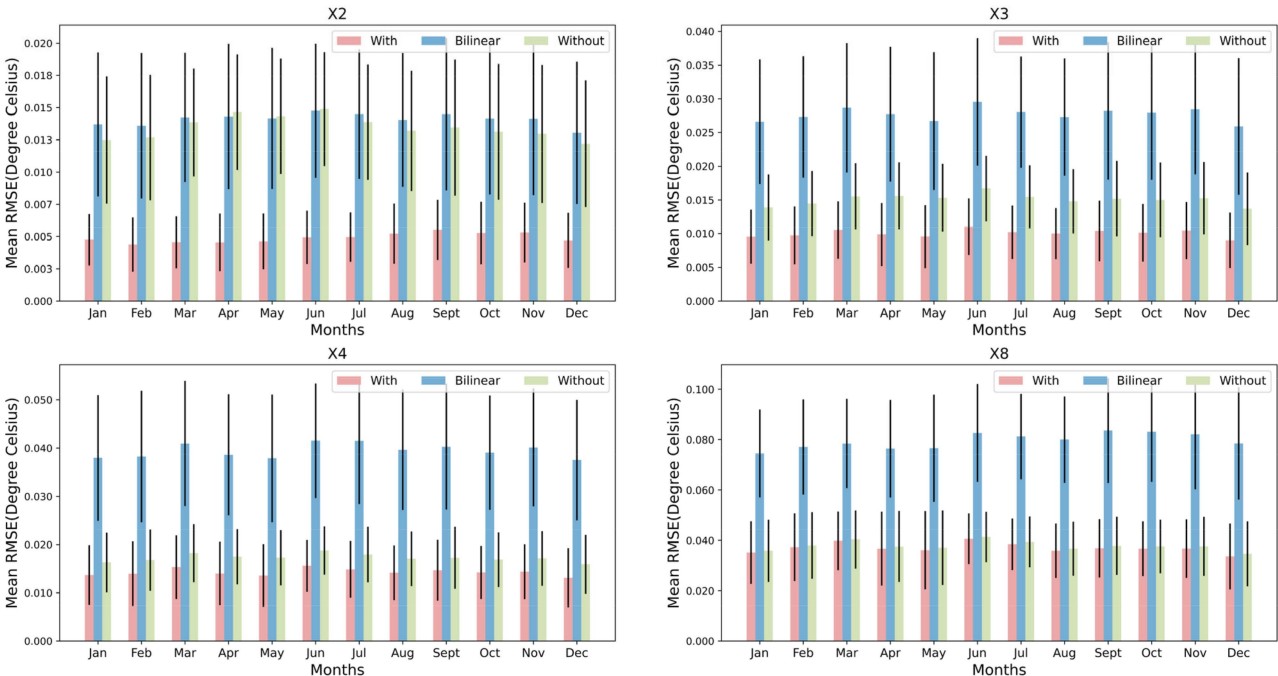

**Figure 8.** Mean error for the proposed interpolation method with and without temporal information under different calendar months and enhancement scales.

To clarify how temporal information affects the results of each sample, we computed the spatial standard deviation for every ground truth. Subsequently, we plotted the relationship between the error of each sample and its corresponding spatial standard deviation across different enhancement scales. Figure 9 shows our results. Across varied enhance-

ment scales, the slope of the trend for results with temporal information is consistently less compared to the one without temporal information. Notably, at enhancement scales of 8 or 12, the trend for results with temporal information exhibits a negative slope. Although the quantitative relationship between errors and spatial standard deviations remains ambiguous due to varying sample sizes corresponding to different spatial standard deviations, a qualitative comparison between RMSE and standard deviation trends suggests that embedding temporal information aids in reducing errors for samples with larger spatial standard deviations.

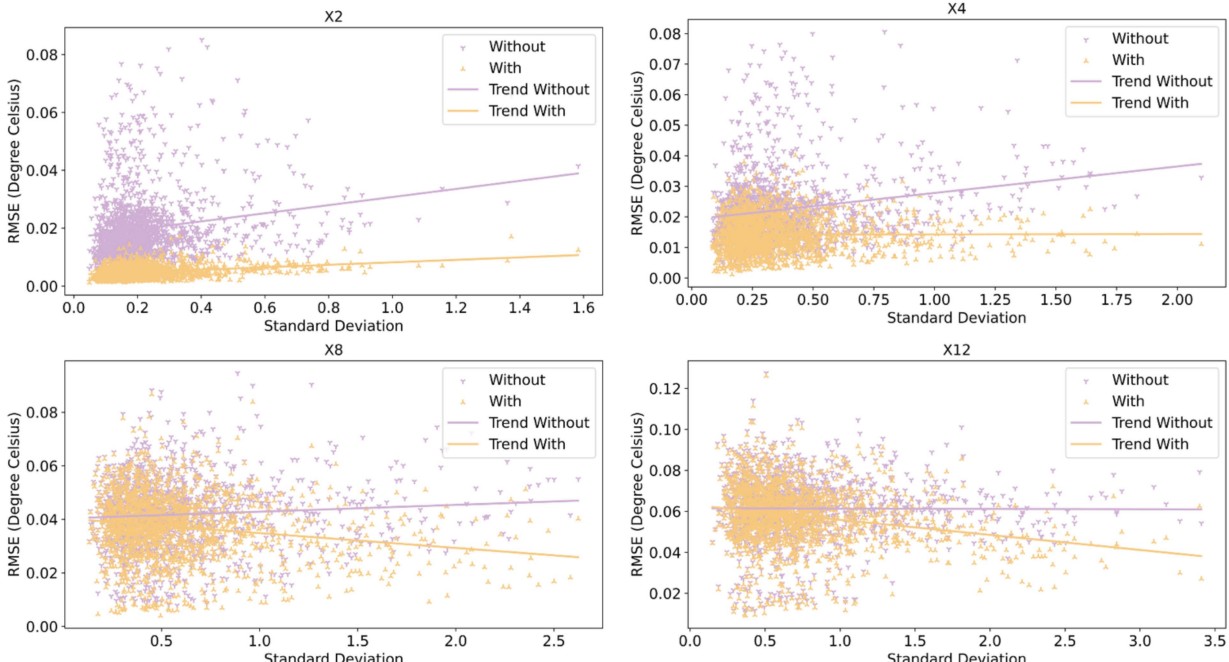

**Figure 9.** Relationship between RMSE and spatial standard deviation of samples, with and without temporal information, across different enhancement scales.

*4.3. Analysis of Using Proposed Method for Recovering Missing Value*

The idea of the proposed interpolation method based on implicit neural representation for obtaining HR results lies in the ability to infer values at unknown locations from known values. In this section, we explore the applicability of the proposed method, with an enhancement scale set to 1, to recover missing values in measurement data. Addressing missing values in remote sensing measurement data holds significant relevance in climate research. For instance, SST derived from infrared sensors can achieve a resolution as fine as 1 km. However, IR-based measurements are susceptible to cloud contamination. Such cloud interference can result in missing values in the data for certain regions. These inherently constrain the utility of the data to some extent. We assessed the efficacy of our proposed method in addressing this challenge.

We employed our pre-trained encoder and MLP for these experiments. Our input data consist of patches from the MUR SST validation dataset, each of size $200 \times 200$. Different proportions of missing data were introduced. Initially, we filled these missing data points with the average value of the known data. Subsequently, these filled patches were input into the encoder to obtain the implicit representation. For each missing data point, we also identified the four nearest known points and, in conjunction with temporal information, utilized the MLP to simulate the value at that position. Results for various proportions can be observed in Table 3. As the proportion of missing data ranges from 5% to 60%, we note that the proposed interpolation method, which embeds temporal information, consistently outperforms other conventional interpolation techniques. For instance, at a 5% missing data ratio, the error for $T\_INRI$ stands at 0.007 °C, slightly superior to

bicubic and linear methods, both registering an error of 0.008 °C. When the missing ratio is at 40%, the error for $T\_INRI$ is 0.011, whereas $INRI$, bicubic, and linear yield errors of 0.014 °C. At a 60% missing ratio, $T\_INRI$ exhibits more stable outcomes. The errors for the method with and without temporal information are 0.012 °C and 0.013 °C, respectively. In contrast, bicubic, linear, and nearest interpolation techniques present errors of 0.017 °C, 0.019 °C, and 0.021 °C, respectively. we also present the standard deviation of the RMSE for different methods at various missing ratios. We can observe that compared to traditional methods, the standard deviation of our $T\_INRI$ under different missing ratios is smaller, approximately half that of the traditional methods. This indicates that the results of our proposed method are more stable. Broadly, $T\_INRI$ demonstrates a pronounced advantage, particularly as the missing data proportion escalates. While the incorporation of temporal information enhances $T\_INRI$'s performance in recovering missing values, the extent of improvement remains modest. The visual outcomes of these experiments can also be seen in Figure 10.

**Table 3.** Comparison of errors corresponding to different methods used to recover missing grid values under different missing data ratios.

| Method | Missing Proportion (RMSE (°C)) | | | | | | |
|---|---|---|---|---|---|---|---|
| | 5% | 10% | 20% | 30% | 40% | 50% | 60% |
| Bicubic | 0.008 ± 0.006 | 0.010 ± 0.007 | 0.013 ± 0.006 | 0.013 ± 0.006 | 0.014 ± 0.007 | 0.014 ± 0.007 | 0.017 ± 0.009 |
| Linear | 0.008 ± 0.006 | 0.008 ± 0.006 | 0.012 ± 0.005 | 0.014 ± 0.006 | 0.014 ± 0.006 | 0.015 ± 0.007 | 0.019 ± 0.008 |
| Nearest | 0.019 ± 0.006 | 0.019 ± 0.006 | 0.019 ± 0.006 | 0.018 ± 0.006 | 0.019 ± 0.006 | 0.020 ± 0.006 | 0.021 ± 0.007 |
| $T\_INRI$ | 0.007 ± 0.003 | 0.008 ± 0.003 | 0.012 ± 0.003 | 0.011 ± 0.003 | 0.011 ± 0.003 | 0.011 ± 0.003 | 0.012 ± 0.004 |
| $INRI$ | 0.008 ± 0.004 | 0.009 ± 0.004 | 0.014 ± 0.004 | 0.013 ± 0.004 | 0.014 ± 0.004 | 0.013 ± 0.004 | 0.013 ± 0.004 |

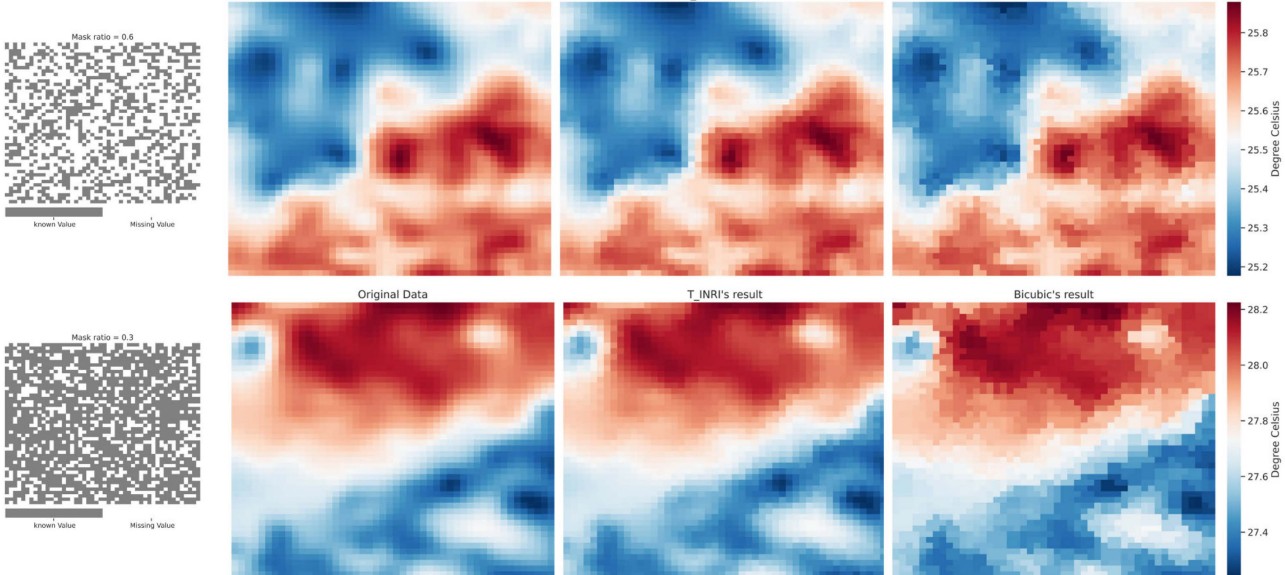

**Figure 10.** Visualization of the original data and results obtained by the proposed interpolation method and bicubic interpolation under different missing ratios. When the ratio is 0.3, the RMSE of the sample for $T\_INRI$ is 0.008, and for the bicubic is 0.014; when the ratio is 0.6, the RMSE of the sample for $T\_INRI$ is 0.015, and for the bicubic is 0.027.

## 5. Conclusions

In this study, we presented $T\_INRI$ designed for generating HR climate data across various enhancement scales. Our method focused on SST data, employing daily HR MUR SST datasets to train the encoder and MLP within $T\_INRI$. For each position in the LR input, the encoder translates it into an implicit representation. To determine values in

unknown HR grids, *T_INRI* pinpoints the closest four known anchors from the LR sample. *T_INRI* garners the information such as implicit neural representations, distances, grid centroid coordinates, and size of the grid. To bolster interpolation accuracy, we incorporate temporal information, specifying the originating calendar month of each LR sample. Using the aggregated information, *T_INRI* employs the MLP to predict the value at an unknown location. The values derived from the four anchors undergo an IDW process for averaging, yielding the final value for the unknown grid. By methodically addressing each unknown grid in HR, *T_INRI* consistently produces corresponding HR outputs from the LR inputs.

Unlike methods such as SRCNN or SRGAN, *T_INRI* strategically harnesses the feature domain to correlate known and unknown positions using distance. This interpolation strategy empowers *T_INRI* to generate HR outputs at arbitrary enhancement scale. In the training process, *T_INRI* employs a self-supervised learning approach. The primary objective is to enable the encoder and MLP to deduce unknown positions using known positions in the feature domain. This training approach ensures that *T_INRI* can not only generate HR data corresponding to enhancement scales encountered during training but also for those scales not directly addressed during the training phase, such as the results at scales of 5, 8, 12, 14, 16, and 20. The findings underscore that *T_INRI* consistently outperforms the alternatives across all enhancement scales.

To elucidate the impact of embedding temporal information, we examined the performance of implicit neural-representation-based interpolation both with and without the inclusion of temporal data. We observed that, across various enhancement scales, the *T_INRI* incorporating temporal information consistently outperformed its counterpart, *INRI*. The superiority of *T_INRI* is particularly pronounced for results corresponding to in-training scales. In the context of out-training-scale results, the enhancement offered by adding temporal information to implicit neural-representation-based interpolation is marginal. By categorizing results from different enhancement scales into distinct calendar months, we determined that the benefits provided by the temporal information are consistent across various months. Incorporating temporal information has the potential to improve HR results for samples with larger spatial standard deviations. Moreover, when applying *T_INRI* to external datasets, we discerned that the embedding of temporal information can enhance the generalization capabilities of generating HR climate data.

As a direction for future research, we plan to explore the use of learnable weights to compute the final unknown grid cell values in *T_INRI*, as opposed to using fixed weights based on inverse distance. In this study, we utilized discrete labels to describe the temporal information of samples. A challenging avenue for future research would be exploring the use of continuous representations to embed temporal information. These endeavors have the potential to further enhance the capabilities of our method and provide new insights into the use of deep learning methods in climate-related research.

**Author Contributions:** Y.W. designed methods, conducted the experiments, and performed the programming. H.A.K. and X.J. revised the paper and supervised the study. All authors have read and agreed to the published version of the manuscript.

**Funding:** This research was funded by the National Science Foundation under Grants IIS-2239175 and IIS-2147195 and NASA under Grant No. 80NSSC22K1164.

**Data Availability Statement:** GHRSST Level 4 MUR Global Foundation Sea Surface Temperature Analysis dataset (MUR SST) can be downloaded from https://podaac.jpl.nasa.gov/dataset/MUR-JPL-L4-GLOB-v4.1 (accessed on 3 May 2023). MUR SST Monthly Mean dataset can be downloaded from https://polarwatch.noaa.gov/erddap/info/jplMURSST41mday/index.html (accessed on 15 May 2023).

**Conflicts of Interest:** The authors declare no conflict of interest.

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
