# Peer review of "Reconstruction of Continuous High-Resolution Sea Surface Temperature Data Using Time-Aware Implicit Neural Representation"

_remotesensing, doi:10.3390/rs15245646_

Round 1

Reviewer 1 Report

Comments and Suggestions for Authors

title: Continuous Super-Resolution of Climate Data Using Time-2 aware Implicit Neural Representation

This paper introduces a novel method for reconstructing high-resolution climate data, focusing on sea surface temperature, which is important for understanding the El Niño-Southern Oscillation's influence on global climate patterns. Unlike traditional deep learning models that are limited to fixed resolution enhancement scales, the proposed temporal information-based implicit neural representation interpolation method (T_INRI) can upscale low-resolution data to any desired resolution. The study shows that incorporating temporal information, such as the specific month of the sea surface temperature data, significantly improves the performance of T_INRI across various scales. Experimental results confirm that T_INRI outperforms conventional methods and is effective at recovering missing data points, making it a valuable tool for climate research and the development of related systems in energy, agriculture, and other sectors.

Author Response

Response:

Thank you for your recognition of our work. We appreciate your feedback and are glad to hear that you found our method valuable for climate research and related applications. Your feedback encourages us to continue our efforts in this field.

Reviewer 2 Report

Comments and Suggestions for Authors

Summary

This paper presents a novel approach to address the fixed scale size issue in climate super-resolution using implicit neural representation techniques. It leverages time-series observations to enhance results. The paper is well-organized, and the results are thoroughly discussed from various perspectives, ensuring the method's applicability across different rescaling factors and addressing the challenge of missing values in climate datasets. The study's focus on Sea Surface Temperature (SST) is particularly noteworthy, as SST is recognized as one of the more challenging climate data types to accurately represent. There are only a couple of minor comments, which are provided in the following bullets.

Minor Comments:

-       Table 1, Page 9: There is no clear line between training results and out-of-training RMSE results. It would be great to separate them by using a vertical line. Also, please provide more details about Table 1, in lines 327-331. Same for Table 2 and so on.

-       Figure 6: There is no definition for Model-WT and Model-WOT. What are they stand for? I assume they are with and without temporal information, would you mind add this information in figure caption and also text.

Author Response

Response:

  • Thank you for your suggestion, we have added vertical lines in both Table 1 and Table 2 to assist readers in better distinguishing the results. For the metrics used in Table 1 and Table 2, we provided an introduction in the Validation section (Root mean squared error (RMSE)). As for the other models utilized in Table 1 and Table 2, we introduced them in the first paragraph of section 4.1. Comparison of Results Generated by Different Models for Arbitrary Scale HR.
  • Your understanding is correct. Model-WT and Model-WOT represent models with and without temporal information, respectively. In the updated version, we have changed 'Model-WT' to T_INRI and 'Model-WOT' to T_INRI in Figure 5 and Figure 6 to maintain consistency with the model representation in the tables and provided explanations in the text.

Reviewer 3 Report

Comments and Suggestions for Authors

In this paper, the authors propose a new interpolation algorithm, namely an implicit neural representation-based interpolation method with temporal information (T_INRI), for reconstructing satellite remote sensing SST data. This method is not limited to fixed-resolution reconstruction as traditional methods. It can enhance low-resolution input by arbitrary scales. In addition, this method also performs higher accuracy than conventional methods in interpolating missing data. The structure and writing are generally good. Although there are still some issues, IMO, I recommend publishing this paper after substantial revisions. My comments and suggestions are as follows:

(1) The title is more or less strange. Its topic is the resolution, not the high-resolution data or the algorithms. Is this true? It would be more appropriate to change it to: Reconstruction of High-resolution Sea Surface Temperature Data Using Time-aware Implicit Neural Representation.

(2) What is the difference between super-resolution and fine-resolution? The Abstract only mentions fine-resolution and high-resolution, not super-resolution. I understand that super-resolution is just a name for SRCNN. Note that the SRCNN method also yields high-resolution products, right? It is not a correct word, at least in this paper. If you must use super-resolution, explain its meaning clearly in the Introduction.

(3) The first two paragraphs of the Introduction talk about ENSO. This is unnecessary, as it is not closely related to the theme. It seems possible to delete them. Note that your method only analyzes SST and is not used for other climate data, though it may also be applicable.

(4) Lines 111-114 should be placed in the Conclusion section.

(5) Figure 1 is just an example. Why choose this day? It's better to use the average of the whole period. Alternatively, provide the amount of data for each grid point, the missing measurement rate of the data, or other valuable parameters. That way, the amount of information is more substantial.

(6) Why is the large gray area in Figure 1 displayed this way? And 'th' clerical error.

(7) The Discussion section did not integrate well with existing research. Now, by comparing the results of different calculation methods, the advantages of the proposed method are demonstrated. However, since the results presented were all provided by the authors, it is necessary to supplement the discussion from the inner aspects of the method, such as explaining the essential differences between your method and conventional methods and why your method is better. When presenting, it is necessary to integrate some existing methods and literature.

(8) The results of Figures 5 and 6 only provide examples for two moments, and the statistics of errors for the whole period should also be supplemented. Please explain what WT and WOT represent, respectively.

(9) Similarly, the results in Figure 10 only have two cases, and some statistically significant results must be supplemented.

(10) In Figure 10, the resulting resolution by the proposed interpolation method seems worse than that by conventional methods.

(11) The Conclusion section is too long, and the conclusion's focus is unclear. Expanding the part based on the two points of Lines 111-114 is better. Three paragraphs are enough, IMO. The first paragraph summarizes the work carried out in this paper and the characteristics of the proposed methods. The second and third paragraphs discuss the two contribution points of Lines 111-114, respectively.

Round 2

Reviewer 3 Report

Comments and Suggestions for Authors

I am glad the authors have fully considered all my comments or suggestions. And I believe that the current version of the paper is good enough to be published.